# A New Mathematical Numerical Model to Evaluate the Risk of Thrombosis in Three Clinical Ventricular Assist Devices

**DOI:** 10.3390/bioengineering9060235

**Published:** 2022-05-27

**Authors:** Yuan Li, Hongyu Wang, Yifeng Xi, Anqiang Sun, Xiaoyan Deng, Zengsheng Chen, Yubo Fan

**Affiliations:** Key Laboratory of Biomechanics and Mechanobiology (Beihang University), Ministry of Education, Beijing Advanced Innovation Center for Biomedical Engineering, School of Biological Science and Medical Engineering, Beihang University, Beijing 100083, China; liyuan951102@163.com (Y.L.); why970127@163.com (H.W.); by2010151@buaa.edu.cn (Y.X.); saq@buaa.edu.cn (A.S.); dengxy1953@buaa.edu.cn (X.D.)

**Keywords:** thrombosis risk model, ventricular assist devices, non-physiological shear stress, residence time, coagulation factors

## Abstract

**(1) Background:** Thrombosis is the main complication in patients supported with ventricular assist devices (VAD). Models that accurately predict the risk of thrombus formation in VADs are still lacking. When VADs are clinically assisted, their complex geometric configuration and high rotating speed inevitably generate complex flow fields and high shear stress. These non-physiological factors can damage blood cells and proteins, release coagulant factors and trigger thrombosis. In this study, a more accurate model for thrombus assessment was constructed by integrating parameters such as shear stress, residence time and coagulant factors, so as to accurately assess the probability of thrombosis in three clinical VADs. **(2) Methods:** A mathematical model was constructed to assess platelet activation and thrombosis within VADs. By solving the transport equation, the influence of various factors such as shear stress, residence time and coagulation factors on platelet activation was considered. The diffusion equation was applied to determine the role of activated platelets and substance deposition on thrombus formation. The momentum equation was introduced to describe the obstruction to blood flow when thrombus is formed, and finally a more comprehensive and accurate model for thrombus assessment in patients with VAD was obtained. Numerical simulations of three clinically VADs (CH-VAD, HVAD and HMII) were performed using this model. The simulation results were compared with experimental data on platelet activation caused by the three VADs. The simulated thrombogenic potential in different regions of MHII was compared with the frequency of thrombosis occurring in the regions in clinic. The regions of high thrombotic risk for HVAD and HMII observed in experiments were compared with the regions predicted by simulation. **(3) Results:** It was found that the percentage of activated platelets within the VAD obtained by solving the thrombosis model developed in this study was in high agreement with the experimental data (r² = 0.984), the likelihood of thrombosis in the regions of the simulation showed excellent correlation with the clinical statistics (r² = 0.994), and the regions of high thrombotic risk predicted by the simulation were consistent with the experimental results. Further study revealed that the three clinical VADs (CH-VAD, HVAD and HMII) were prone to thrombus formation in the inner side of the secondary flow passage, the clearance between cone and impeller, and the corner region of the inlet pipe, respectively. The risk of platelet activation and thrombus formation for the three VADs was low to high for CH-VAD, HVAD, and HM II, respectively. **(4) Conclusions:** In this study, a more comprehensive and accurate thrombosis model was constructed by combining parameters such as shear stress, residence time, and coagulation factors. Simulation results of thrombotic risk received with this model showed excellent correlation with experimental and clinical data. It is important for determining the degree of platelet activation in VAD and identifying regions prone to thrombus formation, as well as guiding the optimal design of VAD and clinical treatment.

## 1. Introduction

Thrombosis is the main complication in patients with clinically placed ventricular assist devices (VAD). The incidence of HMII thrombosis has been reported to be 8.4–11% [1,2,3] and 8.1% for HVAD (0.081 per patient per year) [3,4,5]. Adverse events associated with thrombus, including ischemic stroke, transient ischemic attack, and peripheral thromboembolism, severely impact patient health and the clinical outcomes of VAD treatment. Thrombus formation is profoundly influenced by local surrounding hemodynamics. Platelets which play an important role in thrombosis and hemostasis, can sense their microenvironment and initiate coagulation cascades under specific mechanical and chemical conditions [6]. The high rotating speed of VAD can generate high non-physiological shear stress (NPSS), and the complex configuration of VAD and the turbulent flow within VAD can create flow stagnation regions. The formation of thrombus is closely related to NPSS [7,8] and flow stagnation. The effect of shear stress on platelet activation and aggregation has been extensively studied. Platelets have long been recognized as a special cell fragment that mediates physiological hemostasis and plays a key role in pathological thrombosis [8]. Platelet dysfunction can affect normal hemostasis processing, which leads to thrombosis or bleeding [9]. The activation of platelets can increase the risk of thrombotic complications [10]. Pathological shear stress levels of 31.5 Pa encountered in atherosclerotic arteries with severe atherosclerosis activate platelets and trigger platelet microparticles production [8]. When NPSS (>100 Pa) is high, platelet activation can be induced even at very short exposure time (<1 s) [7,11]. Residence time is another major cause for thrombosis formation. Studies have shown that low-level shear stress (<0.35 Pa) with a long residence time [11] can lead to the release of small amounts of ATP, ADP, and 5-hydroxytryptamine and further cause platelet aggregation [10,12]. At the same time, the deposition of blood components (platelets, fibrinogen, etc.) in the regions with long residence time can also lead to the formation of thrombosis [13,14].

Because of the complicated structure of VAD, it is hard to observe or investigate the thrombosis formation within VAD. It is useful to use the computational fluid dynamics (CFD) method to build the thrombosis model for understanding thrombosis formation and assessing thrombotic risk within the VADs. The thrombus model associated with the VADs requires the incorporation of complex flow characteristics within the VADs, including high shear stress and long residence times caused by the turbulence structure, complex geometries, and high rotating speeds. In addition, thrombus models should be implementable in computational CFD solvers, and ideally, such models should be compatible with commercial CFD software for broader adoption in the scientific and engineering communities [15]. Several researchers have assessed thrombotic risk by Lagrangian methods [16,17,18,19,20,21]. In their studies, particles representing platelets are released at the inlet of the studied subjects and the trajectories of platelets movement are continuously tracked. The information on the platelet trajectories, such as stress accumulation and residence time, are employed to evaluate the likelihood of thrombosis in the study subjects. However, this method can only qualitatively assess the likelihood of thrombosis in different study subjects, which cannot identify the location of high thrombotic risk. Additionally, other thrombus prediction methods based on volume of fluid (VOF) models have also been proposed [22,23,24,25,26]. That is, a moment in time is chosen as the fully developed flow field. After this moment, the fluid at the inlet is defined as a new phase (new blood), which has the same physical properties as the fluid already in the computational domain (old blood). By solving for the variation of the flow field of the two phases (new blood and old blood) with time, it is possible to obtain the volume fractions of the new and old blood phases at different moments, and subsequently the location of the residual old blood, as a way to identify regions of high thrombotic risk. However, this method requires significant computational resources, relies excessively on artificially set thresholds, and is unable to predict thrombosis formation induced by high shear stress. Furthermore, some chemical reactions based on the mathematical model, describing the chemical reactions between multiple substances such as resting platelets, activated platelets and coagulation factors by additional convective diffusion equations, were proposed. For example, Xu et al. [27] considered the interaction between residence time, activated platelets and thrombus clots; Taylor et al. [28] considered the interaction between resting platelets, activated platelets and adenosine diphosphate (ADP). However, those models had not been applied in more complex flow geometries such as VADs. Furthermore, this models have been developed for laminar flow and resolves the formation time of a macroscopic thrombus which was around 30 min [15]. This proves to be impractical for the simulation of VAD where a high time resolution is required for adequate simulation accuracy. Wu et al. [29] proposed a thrombotic model involving interactions between a dozen coagulation factors and platelets. This model was applied in a clinical VAD (HMII). However, the simulation is computationally expensive and has limited predictive power. More specifically, thrombus formation was predicted only in one region (front guide vanes) and not in other known/common regions such as the impeller or further downstream locations such as diffuser. Christopher et al. [15] developed a simple thrombotic model applicable to VAD based on the study of Talyor et al. [28]. Their model describes the interaction between resting platelets, activated platelets and ADP through three transport equations and applied the model to HMII. However, this model can only qualitatively predict the thrombotic risk in different regions of HMII, but cannot precisely obtain data such as the percentage of active platelets and does not take into account the flow obstruction caused by thrombosis generation. In addition, the simulation method for predicting thrombus requires a two-stage process. In the first stage, basic physical fields such as velocity and pressure are calculated. In the second stage, the thrombus risk is simulated by the constructed transport equations. This repetitive simulation also causes an increase in computational cost.

In this study, a more comprehensive and accurate thrombosis model was constructed by combining parameters such as shear stress, residence time, and coagulation factors. A mathematical model was employed to assess platelet activation and thrombosis within the VAD. By solving the transport equation, the influence of various factors such as shear stress, residence time and coagulation factors on platelet activation was considered. The diffusion equation was applied to determine the role of activated platelets and substance deposition on thrombus formation. The momentum equation was also introduced to describe the obstruction to blood flow when thrombus is formed. Synchronous simulation of the fundamental physical field and thrombus model was achieved by constructing subdomains. Numerical simulations were performed for evaluating the thrombosis risk in three clinically VADs (CH-VAD, HVAD and HMII) using this model. The simulation results were compared with experimental results on platelet activation caused by these three VADs. The simulated thrombogenic potential in different regions of MHII was compared with the frequency of thrombosis occurrence in the regions in clinic. The regions of high thrombotic risk for HVAD and HMII observed in experiments were compared with the regions predicted by simulation.

## 2. Materials and Methods 

### 2.1. Studied VADs and Geometries

The CH-VAD (CH Biomedical, Inc., Suzhou, China) is a fully magnetically levitated centrifugal pump. The flow characteristics of the CH-VAD include a main flow passage from the axial inlet to the tangential outlet and a U-shaped secondary flow passage around the impeller in the maglev clearance between the rotor and the back volute (Figure 1a). A guide cone is placed to introduce the axial main flow from the inlet into the impeller passage. The secondary flow in the CH-VAD re-enters the impeller passage at the middle portion and meets the main flow inside the impeller passage perpendicularly. The clearance size of the secondary flow passage is approximately 0.25 mm [24,26,30]. 

The HVAD (Medtronic/HeartWare, Framingham, MA, USA) is a centrifugal blood pump featuring a four-blade impeller levitated by a hybrid hydrodynamic and magnetic bearing system (Figure 1b). A secondary flow passage exists in the clearance between the impeller and the back volute and the cone. The cone directs the main flow from the axial inlet to the radial impeller passage. Different from the CH-VAD, the secondary flow meets and mixes with the main flow prior to the blade leading edges [24,31]. 

The HM II (Abbott, Pleasanton, CA, USA) is an axial flow pump with an impeller supported by two ball-cup mechanical bearings (Figure 1c) connected to front guide vanes and the diffuser [24,26]. 

The geometries of the three studied VADs were obtained from computer-aided drawing (CAD) files or constructed by measuring the actual device components through a reverse engineering procedure.

### 2.2. Residence Time and Non-physiological Shear Stress

Residence time (RT) and non-physiological shear stress (NPSS) are thought to play an important role in the activation of platelets and thrombus formation. 

RT is modeled as a tracer of passive transport with blood flow and obeys the following equation [27]:(1)$$\frac{\partial \mathrm{RT}}{\partial t}+v\cdot \nabla \mathrm{RT}=D_{\mathrm{RT}}\nabla^{2}\mathrm{RT}+1$$
where t is time, v is the velocity of the blood, and D_RT_ represents the self-diffusivity of the blood (D_RT_ = 1.14 × 10^−11^ m^2^/s [27]), and the source term is defined as 1/s, that is, the source is consistent with the variation of time.

The viscous scalar shear stress (SSS) was derived from the simulated flow fields to represent NPSS according to the following formula [32]:(2)$$\sigma ={\left[\frac{1}{6}\sum {\left(\sigma_{\mathrm{ii}}-\sigma_{\mathrm{jj}}\right)}^{2}+\sum \left(\sigma_{\mathrm{ij}}\sigma_{\mathrm{ij}}\right)\right]}^{\frac{1}{2}}$$
where the SSS was calculated by multiplying the shear rate tensor $\sigma_{\mathrm{ij}}=\partial v_{i}/\partial x$, with the blood viscosity, μ.

Since it had been proved that it creates numerically elevated shear fields for including the Reynolds stress tensor into the shear stress calculation [33,34], the components of the shear stress in Equation (2) are obtained by considering the viscous stress tensor only in the present study [34,35]. 

### 2.3. Activated Platelets and Thrombotic Clots Risk

#### 2.3.1. Resting Platelets, Activated Platelets and Coagulant Factors

Resting platelets (RPs), activated platelets (APs) and coagulation factors (CFs) were modeled as dilution chemicals passively transported with blood flow. In this model, platelet-activating and thrombus-inducing factors (such as ADP, TXA2, thrombin etc.) are not specifically categorized and are collectively referred to as CFs. This abstraction contributes to the reduction of uncertainty and the improvement of computational efficiency of numerical simulations. RPs are activated by NPSS to form APs, and APs release CFs, which further facilitate the conversion of RPs to APs. APs and CFs tend to accumulate in the flow stagnation and re-circulation regions, that is, in the long residence time (RT above 1 s) region [27,36].The transport equation was employed to describe the above process:(3)$$\frac{\partial \mathrm{RPs}}{\partial t}+v\cdot \nabla \mathrm{RPs}=D_{p}\nabla^{2}\mathrm{RPs}-\left(k_{1}\varphi_{\mathrm{NPSS}}\mathrm{RPS}+k_{2}\varphi_{\mathrm{CTs}}\mathrm{APs}\right)$$
(4)$$\frac{\partial \mathrm{APs}}{\partial t}+v\cdot \nabla \mathrm{APs}=D_{p}\nabla^{2}\mathrm{APs}+k_{1}\varphi_{\mathrm{NPSS}}\mathrm{RPs}+k_{2}\varphi_{\mathrm{CTs}}\mathrm{APs}$$
(5)$$\frac{\partial \mathrm{CTs}}{\partial t}=D_{\mathrm{ceff}}\nabla^{2}\mathrm{CFs}+k_{\mathrm{CFs}}\mathrm{APs}$$
where k_1_ and k_2_ are the user-defined coefficients 0.3/s and 0.1/s, respectively. And φ_CTs_ and φ_NPSS_ are functions of the variables and their thresholds. k_CFs_ is 3 × 10^−17^ mol [28], represents the concentration of CFs released from one APs. v is velocity, m/s. D_p_ and D_ceff_ is the convective diffusion coefficient, defined as a function of shear strain rate, γ [27]:(6)$$D_{p}=D_{\mathrm{pt}}+\alpha \gamma =D_{\mathrm{pt}}+\frac{\alpha \cdot \mathrm{NPSS}}{\mu }$$
(7)$$D_{\mathrm{ceff}}=D_{c}\varphi_{\gamma }$$
where D_pt_ = 1.6 × 10^−13^ m^2^/s [37], α = 7×10^−13^ m^2^ [37], D_c_ = 10^−8^ m^2^/s [27], φ_γ_ are functions of the variables and their thresholds and μ is blood viscosity.

#### 2.3.2. Thrombotic Clots and Thrombosis Obstruct Flow

Thrombotic clots (TCs) are thought to form in regions with high APs concentrations and long residence times. The high concentration of CFs released from regions with high APs concentrations exacerbates platelet aggregation. The deposition of blood components (platelets, fibrinogen, etc.) in the areas with long RT can lead to the formation of thrombosis [13,14]. In this model, TCs is considered to have no diffusion capacity [27], so the diffusion term is 0:(8)$$\frac{\partial \mathrm{TCs}}{\partial t}=k_{\mathrm{TCs}}\varphi_{\mathrm{TCs}}\frac{\mathrm{APs}}{\mathrm{APs}+\mathrm{BPs}}+k_{\mathrm{TCs}}\varphi_{\mathrm{RT}}$$
where k_TCs_ is the source term coefficient, 2 × 10^−2^ mol/s [38], and φ_TCs_ are functions of the variables and their thresholds.

The Navier-Stokes equations were employed to describe the obstruction of blood flow by TCs, and a negative source was added to model a fictitious force in the region of thrombus growth against the fluid motion. The adapted equation is reported below [39]:(9)$$\rho \left(\frac{\partial v}{\partial t}+v\nabla v\right)=-\nabla P+\mu \nabla v-k_{M}\varphi_{\mathrm{TCs}}$$
where k_M_ is the source term coefficient, 10^7^ kg/(m^3^s) [27], φ_TCs_ are functions of the variables and their thresholds, and μ is a blood viscosity that takes into account the thrombus viscosity, which is allowed to change to simulate the increase in resistance in the region where the thrombus is present [40]:(10)$$\mu =\mu_{0}\;\left(1+100\varphi_{\mathrm{TCs}}\right)$$

And μ_0_ is 0.0035 Pa·s [41].

In the present study, the TCs risk data were normalized for better comparison with experimental/clinical data. The employed normalization method was max-min normalization with the mathematical expression, showing as below:(11)$$x_{n}=\frac{x_{o}-\min\left(x_{o}\right)}{\max\left(x_{o}\right)-\min(x_{o})}$$
where x_n_ is the normalized data and x_o_ is the original data.

#### 2.3.3. Thresholds and Initialization

φ_i_ is a function of the variable i and its threshold i_0_, defined as:(12)$$\varphi_{i}=\frac{i}{i_{0}}$$

The shear rate threshold of 10,000/s (35 Pa for NPSS) proposed by Xu et al. [27] was employed as the threshold value for this study. Based on the fitted NPSS-RT curve by Fraser et al. [42] (Figure 2), the threshold value of the residence time was obtained as 1 s (red marker), which showed that platelets can be activated induced by shear stress greater than or equal to 35 Pa for a maximum exposure time of 1 s. Whereas, if shear stress is less than 35 Pa, the role of shear-indued platelet activation leading to thrombotic risk decreases due to the setting of the threshold function of Equation (12), and the long residence time leading to blood pooling is the main factor for thrombus formation. Therefore, one second was applied as the threshold value for blood pooling. When the variable is below 2 to 3 times the threshold, then the reaction resulting from that variable can be considered to be completely stopped [27]. The thresholds for the other variables and their origins can be found in Table 1.

The inlet, outlet, and wall boundary conditions regarding the reaction variables can be found in Table 2. For the transport of CFs, a flux boundary condition (Flux 1) depending on shear stress and TCs concentration was employed to describe the release of CFs at low shear on the wall when TCs did not reach the threshold, with the following expressions [38]:(13)$$D_{\mathrm{ceff}}\frac{\partial \mathrm{CFs}}{\partial n}=\{\begin{array}{c} k_{\mathrm{ceff}}\;\mathrm{if}\;\mathrm{Shear}\;\mathrm{Stress}<0.2\mathrm{Pa}\;\mathrm{and}\;\mathrm{TC}\;\leq \;200\mathrm{nmol}\\ \;0\;\mathrm{otherwise}\; \end{array}$$

### 2.4. CFD Methods

The blood in the VAD is considered to be a three-dimensional, incompressible Newtonian fluid [20,48,49] with a density of 1055 kg/m^3^ and a viscosity of 0.0035 Pa·s [41]. The rotating regions and the stagnant regions of studied VADs are combined by the frozen rotor interface [42,50,51]. The Reynolds-averaged Navier-Stokes (RANS) equations were solved using the commercial software ANSYS CFX 18.2 (ANSYS Inc., Canonsburg, PA, USA), which employs a finite-volume method based discretization of the governing equations. The convective terms were solved in high-resolution form and the SST k-ω turbulence model was employed for steady simulations [41,51]. A mass flow rate of 0.079 kg/s (obtained from the volume flow rate of 4.5 L/min) boundary condition was set at the inlet and a pressure boundary condition (0 mmHg) was set at the outlet. To allow full development of flow, the inlet and outlet of the VADs were extended (approximately 10 times larger than the diameter of the inlet or outlet). To ensure the reliability of the numerical conclusions, the risk of platelet activation and thrombosis in the extended segments was not counted. All solid walls were assumed to be no-slip and adiabatic, and the convergence criterion was set to be 10^−6^. 

The source terms for equations (1–10) are implemented in the CEL language of the commercial software ANSYS CFX 18.2 (ANSYS Inc., Canonsburg, PA, USA). The source terms for these user-defined variables are integrated into subdomains and solved synchronously with the underlying physical quantities (such as, pressure, velocity, etc.) by calling mass and momentum conservation equations. APs at the VAD outlet and the volume-averaged CTs across the VAD domain were extracted to determine the degree of platelet activation by VAD and the potential for thrombus formation.

During the simulation, all these three pump speeds were chosen according to their H-Q curve (3000 rpm for CHVAD, 2800 rpm for HVAD and 9200 rpm for HMII, respectively), which can make sure that the pressure heads of them are very close (around 70 mmHg) under the flow rate of 4.5 L/min [26]. The calculations were performed on a PC workstation with two Intel Glod 6354 (18 cores, 3 GHz) processors. Twenty threads were allocated to each calculation, and it took about 8 h to converge.

### 2.5. Mesh Details and Sensitivity Analysis

ANSYS ICEM (ANSYS Inc., Canonsburg, PA, USA) was employed to generate a hybrid mesh and a local mesh refinement of the clearance (Figure 3). The mean size of the domain mesh is set to be 0.3 mm and the boundary layer mesh size is 0.02 mm, making 6.5 million, 8.4 million and 3.6 million meshes (medium mesh, M) for CH-VAD, HVAD and HM II, respectively, on which the amount of meshes is reduced and doubled respectively to form a coarse mesh (C) and a fine mesh (F) for mesh independent verification. The mesh independence was verified at its clinic condition and can be seen that the densities of the medium and fine meshes differ very little in terms of pressure head (about 1.5%), percentage of APs (about 1.5%), and volume average TCs (about 2.0%). Therefore, the medium mesh was chosen as a trade-off between accuracy and wall-clock time.

### 2.6. Rotor Position Independent Verification

Since steady simulations are employed in this study, different relative positions of rotor and volute tongue may result in different flow field. Therefore, the effect of rotor position on the results of this study needs to be investigated. For the two centrifugal VADs, three cases with the trailing edge of the rotor blade facing the volute tongue and ±20° away from the tongue were investigated. For the axial VAD, the case where the trailing edge of the rotor blade is aligned with the leading edge of the diffuser vane and two cases where it is ±20° off are investigated.

For HVAD and HMII, the change of rotor position has almost no effect on the flow field. For CH-VAD, the variation in the relative position of the rotor and the volute tongue hardly affects the flow field in the rotor region, but has a little effect on the flow field near the volute tongue (Figure 4). This effect is manifested in the different location and size of the flow separation in the volute tongue region/diffusion pipe. However, this difference arises more from the limitations of the steady simulation method. This is due to the steady simulation may not capture the flow in the volute tongue region accurately [52], which did affect our thrombosis risk evaluation because this flow separation can be captured for different rotor positions. Despite the bias, it is acceptable for the study. Furthermore, the relative positions of the rotor and volute tongue were different, and the obtained pressure head, percentage of platelet activation and risk of TCs were hardly consistent, also proving that the rotor position in this study caused little effect.

## 3. Results

### 3.1. Hemodynamic Results and Simulation Results Validation

The studied VADs were all able to provide pressure heads about 70 mmHg under their clinical conditions [24,26,41]. The simulation results were in good agreement with the experimental results [26] (mean error of 2%), demonstrating the accuracy of the numerical method (Figure 5a). The activated platelets by the three VADs obtained by simulation under simulation conditions (Figure 5b) was in highly consistent with the experimental results performed by Zachary et al. [26] with healthy human blood at the same conditions (correlation coefficient r^2^ = 0.984). This proves the accuracy of the platelet activation model developed in this study. Among the three studied VADs, CH-VAD had the lowest to activate platelets and the lowest probability of thrombus appearance and the HMII had the highest possibility of platelet activation and thrombosis formation, as reflected by the normalized thrombosis risk (based on volume-averaged TCs) (Figure 5c). The present study also compared the risk of thrombosis in three regions within the HMII (Figure 1c) with the risk of thrombosis in these three regions based on the clinical statistics of the HMII performed by Rowlands et al. [45] They carried out a retrospective analysis on 29 HMII devices that had thrombus after clinical use to determine the frequency, composition, and localization of macroscopic thrombosis. The high correlation between normalized simulations and statistical results (Figure 5d) demonstrates the validity of the thrombus model constructed in this study (correlation coefficient r^2^ = 0.994).

### 3.2. Velocity Fields

For the centrifugal VADs (Figure 6a,b), the incoming flow is guided by the cone and its flow direction changes from axial to radial into the impeller passage of the VADs. The blood accelerates in the impeller and enters the volute. Due to the diffusion of the volute, the blood flow in the volute slows down and a part of the kinetic energy is converted into pressure energy. At the same time, driven by the pressure difference between the volute and the impeller, a part of the blood moves back and re-enters the impeller passage through the top clearance, back clearance, and the clearance between the impeller and the cone. For the axial VAD (Figure 6c), the blood first moves axially through the front guide vanes, then blood flow is accelerated in the impeller passage and finally enters the downstream diffuser. In the diffuser, the blood is slowed down and a portion of the kinetic energy is converted to pressure energy.

### 3.3. Scalar Shear Stress and Residence Time

Platelets are activated in regions of high scalar shear stress (HSSS) [6,54], and the regions of long residence time (LRT) lead to deposition of blood components [13,14]. The combined effect of HSSS and LRT increasing the risk of thrombosis formation. Therefore, identification of the HSSS and LRT regions within the VAD is important for localizing the regions of platelet activation and thrombosis formation as well as determining the level of platelet activation and the probability of thrombotic risk. According to Equation (12), when the variable is less than 2 to 3 times the threshold value, the reaction in which the variable is involved can be considered to be completely stopped [27]. Therefore, the region of three studied VADs with shear stress greater than 15 Pa (HSSS) and residence time greater than 0.35 s (LRT) is shown in Figure 7.

For CH-VAD, influenced by the pressure difference between the volute and the impeller, the high-speed blood from the impeller flows in reverse into the narrow top clearance and secondary flow passage (Figure 6a, marker 1), so HSSS appears in the secondary flow passage (Figure 7a, marker 1) and the top clearance (Figure 7a, marker 2). The velocity of the blood in the inlet pipe is low on both sides and high in the middle due to the effect of the viscous resistance, and the LRT appears near the wall (Figure 8a, marker 1). As the blood flow moves in the secondary flow passage, its kinetic energy decreases, so the velocity of the inner side of the secondary flow passage is much lower than that of the outer side (Figure 6a, marker 1), resulting in LRT in the inner side of the secondary flow passage (Figure 8a, marker 1). Also, the collision of blood from the secondary flow passage with the main flow causes significant energy damage and interferes with the flow of the main flow (Figure 6a, marker 2), leading to the appearance of HSSS (Figure 7a, marker 3) and LRT (Figure 8a, marker 3). The large pressure difference between the two sides of the volute tongue also leads to the occurrence of unstable flow near the volute tongue (Figure 6a, marker 3), which results in the appearance of LRT (Figure 8a, marker 4). 

Due to the influence of the viscous resistance, LRT also exists in the near-wall region of the inlet pipe of HVAD (Figure 8b, marker 1). Since HVAD is a hydrodynamic suspension VAD, it has a narrower secondary flow passage (hydrodynamic clearance) in which the secondary flow results in a larger HSSS (Figure 7b, marker 1 and 2). The thick blades of HVAD reduce the effective overflow area of the impeller. The blood from the secondary flow passage is circulated and continuously impinged in the clearance between the cone and the impeller (Figure 6b, marker 1), leading to the generation of HSSS (Figure 7b, marker 3) and LRT (Figure 8b, marker 2). The high velocity blood from the hydraulic clearance and the disordered blood from the clearance between the cone and impeller are mixed with the main blood flow in the impeller passage (Figure 6b, marker 2), leading to the appearance of HSSS (Figure 7b, marker 4). 

For HMII, the sudden variation in the direction of blood flow causes a large vortex region to appear in the region of inlet pipe corners (ahead of the front guide vanes) (Figure 6c, marker 1). The stagnation and re-circulation of blood in the region leads to the appearance of LRT (Figure 8c, marker 1). The high rotating speed of the rotor causes the pressure in the rotor region to be much higher than the pressure in the region of the front guide vanes, impeding some of the blood flow. The part of blood flow is continuously recirculated in the region of front guide vanes (Figure 6c, marker 2), leading to the appearance of LRT in the region (Figure 8c, marker 2). At the same time, the blood is accelerated in the rotor domain. In particular, the blood velocity increases significantly when blood enters the narrow blade top clearance (Figure 6c). This results in the rotor region with a high velocity gradient and HSSS (Figure 7c, market 1). This high velocity blood continues to flow into the downstream diffuser (Figure 6c), creating HSSS in the diffuser region (Figure 7c, marker 2 and 3). At the same time, the spatially curved blades of diffuser convert the kinetic energy of the blood into pressure energy. The velocity in the diffuser decreases significantly, especially at the end of the diffuser (Figure 6c). As a result, the residence time of blood flow within and after the diffuser increases (Figure 8c). Additionally, the existence of the diffuser blades also impedes blood flow, and the blood flow may collide with the diffuser blades. Therefore, significant vortices emerge in the diffuser region (Figure 6c, marker 3). These recirculated fluids also lead to the appearance of local LRT (Figure 8c, marker 3).

### 3.4. Platelet Activation Assessment

Activated platelets (APs) are appeared in the HSSS region [6,54] and tend to be deposited in the LRT regions [27,36].

For CH-VAD, the LRT in the near-wall region of the inlet pipe leads to a higher deposition of APs (Figure 9a, marker 1). The HSSS in the secondary flow passage greatly activates platelets, and APs accumulate in the inner side of the secondary flow passage (HSSS and LRT region) as the blood flow moved (Figure 9a, marker 2). The combination of HSSS (generated by the collision of secondary flow with the main flow) and the LRT (generated by the disturbance of secondary flow to the main flow) results in a higher number of APs appearing near the intersection region of the secondary flow and the main flow (Figure 9a, marker 3). The LRT region near the volute tongue leads to more APs stagnation here (Figure 9a, marker 4).

For HVAD, the LRT in the near-wall region of the inlet pipe leads to the pooling of APs in this region (Figure 9b, marker 1). The HSSS generated in the hydraulic clearance greatly activates the platelets, resulting in a high percentage of APs in the clearance (Figure 9b, marker 2 and 3). The unstable secondary flow in the clearance between the cone and impeller generates HSSS and LRT, which also lead to a high percentage of APs occurred (Figure 9b, marker 4). The HSSS and APs in the hydraulic clearance enters the impeller passage, leading to the further activation of platelets in the impeller passage (Figure 9b, marker 5).

For HMII, an apparently high concentration of APs appeared at the corner regions of the inlet pipe, that is ahead of the front guide vanes (Figure 9c, marker 1), due to the LRT. The re-circulation flow in the front guide vanes region also leads to a higher number of APs stagnating here (Figure 9c, marker 2). The HSSS generated by the narrow blade top clearance greatly activates the platelets (Figure 9c, marker 3). The deceleration and pressurization effect and the obstruction of blood flow of the diffuser leads to the presence of HSSS and LRT in this region, which also cause occurrence of more APs. (Figure 9c, marker 4).

Among the three VADs, HMII showed the greatest proportion of APs (21.95%) (Figure 5b) due to the severe LRT at its inlet pipe corner (Figure 9c). HVAD had the middle percentage of platelet activation among the three VADs studied (14.90%) (Figure 5b). The proportion of APs was greatest at its hydraulic clearance and at the region of the clearance between the cone and impeller (Figure 9b). CH-VAD had the lowest proportion of APs among the studied VADs (5.95%) (Figure 5b). In CH-VAD, high percentage of APs were found mainly in and around the secondary flow passage (Figure 9a).

### 3.5. Thrombosis Risk Assessment

In the thrombotic model proposed in this study, regions with higher percentage of APs and longer residence times are considered to have a higher potential for thrombosis. Among the three studied VADs, the risk of thrombosis was in descending order: CH-VAD, HVAD and HM II (Figure 5c), consistent with the variation in the proportion of APs (Figure 5b).

The regions of high thrombotic risk for CH-VAD are mainly found in the inner side of the secondary flow passage (Figure 10a, marker 1), where the proportion of APs is higher (Figure 9a marker 2) and the residence time is longer (Figure 8a marker 2). 

Regions of high thrombotic risk in HVAD are identified mainly in the clearance between cone and impeller (Figure 10b, marker 1), and then in the hydraulic clearance (Figure 10b, marker 1 and 2). The regions predicted by our CFD simulation are the same as the regions of thrombus formation photographed by Schalit et al. [4] employing porcine blood under the same clinical supported conditions. 

Regions of high thrombotic risk in HM II are thought to appear in the corner region of the inlet pipe due to long residence time and high platelet concentrations (Figure 10c, marker 1). Rowlands et al. [53] counted the frequency of clinically used HM II with thrombus in the three regions of the front guide vanes, rotor and diffuser (Figure 10c). Normalized results of Rowlands et al. [53] is consistent with the normalized results of the amount of substance of TCs in these three regions obtained from the numerical simulations of this study (Figure 5d). Antaki et al. [55] tested HMII with bovine blood under the similar clinical conditions as in this study. The variation of TCs with time captured by the experiment is reproduced in Figure 10e. It can be found that the regions of high thrombotic risk obtained by experiment is in high agreement with our CFD simulation results. The thrombus first appeared in the corner regions of the inlet pipe (marker 1 in Figure 10c,e) and was then found at the bottom regions of the front guide vanes (marker 2 in Figure 10c and 10e). With time delay, the thrombus starts to appear in the region of the diffuser inlet (marker 3 and 4 in Figure 10c,e).

## 4. Discussion

Thrombosis is the most significant complication in patients with clinically placed VAD. Models that accurately predict the risk of thrombus formation in VADs are still lacking. When VADs are clinically assisted, their complex geometric configuration and high rotating speed inevitably generate complex flow fields, high shear stress and long residence time. These non-physiological factors can damage blood cells and proteins, release coagulant factors and trigger thrombosis [56,57,58]. The long retention time can lead to the deposition of substances in blood flow field, such as platelets, fibrinogen, further exacerbating the risk of thrombosis [13,14]. In our thrombus model, resting platelets, activated platelets and coagulation factors were modeled as dilution chemicals passively transported with blood flow. Resting platelets are activated by NPSS to form activated platelets, and activated platelets can release coagulation factors, which further facilitate the conversion of resting platelets to coagulation factors. Coagulation factors and activated platelets tend to accumulate in the flow stagnation and re-circulation regions. Regions of high platelet concentration and prolonged residence time are considered to be regions prone to thrombosis. Also, the momentum equation was introduced to model the obstructive effect on local flow after the formation of thrombus. To improve computational efficiency, subdomains were created to enable simultaneous simulation of the fundamental physical fields, platelet activation and thrombus formation. It was found that the percentage of activated platelets within the VAD obtained by solving the thrombosis model developed in this study was in high agreement with the experimental data (r² = 0.984) (Figure 5b), the simulated risk of thrombosis in different regions of HMII showed excellent correlation with the clinical statistics (r² = 0.994) (Figure 5d), and the regions of high thrombotic risk predicted by the simulation were consistent with the experimental results (Figure 10). Further study revealed that the three clinical VADs (CH-VAD, HVAD and HMII) were prone to thrombus formation in the inner side of the secondary flow passage, the clearance between cone and impeller, and the corner region of the inlet pipe, respectively (Figure 10). The risk of platelet activation and thrombus formation for the three VADs was low to high for CH-VAD, HVAD, and HMII, respectively (Figure 5).

The thrombotic model constructed in this study was developed base on the work of Xu et al. [27] and Christopher et al. [15]. The concept of platelet activation by shear stress leading to thrombus formation was introduced based on the thrombus model of Xu et al. [27] The thrombus formation and the effect of thrombus on the flow field were further described based on the thrombus model of Christopher et al. [15]. In the thrombus model constructed in this study, thrombus formation is caused by two main factors, one is caused by shear stress-induced platelet activation and the other is caused by blood stagnation. The thrombus formation described in the model of Xu et al. [27] is mainly dependent on the long residence time region. Therefore, thrombus clots in the hydraulic clearance of HVAD cannot be effectively predicted if adopting the model of Xu et al. [27]. Previous studies have demonstrated that thrombus in the hydraulic clearance of HVAD is white thrombus (thrombus due to platelet activation) [55]. The model of Christopher et al. [15] adopts platelet concentrations to represent regions of high thrombotic risk. While, their study did not consider the further effects on blood flow after thrombosis formation. Furthermore, the study of Christopher et al. [15] only obtained a qualitative concentration of activated platelets, which is difficult to be validated precisely with experiments. This study built an improved thrombus model base on the work of Xu et al. [27] and Christopher et al. [15]. The obtained platelet activation percentage was accurately corresponded with experiments (Figure 5). Also, the predicted thrombotic risk regions based on platelet activation and blood stagnation were in high agreement with experimental or clinical observations (Figure 10).

The sensitivity of the parameters to platelet activation percentage and thrombotic risk was also investigated in this study. Variations in the coefficients k_1_ and k_2_ in Equations (3) and (4) were found to have a significant effect on the same VAD obtained for activated platelets, but did not modify the results of the comparison between different VADs. The coefficients k_CFs_ and k_TCs_ in Equations (5) and (8) and the thresholds only affect the rate of reaction of platelets to activation and thrombus to formation. Regions of high activated platelet concentration and high thrombotic risk do not vary with these coefficients and the thresholds. In fact, thrombus formation is profoundly influenced by the flow field, and the high shear stress and long residence time due to flow are the underlying causes of thrombus formation.

Based on the above, the thrombotic risk model proposed in this study can accurately assess the proportion of platelet activation and the probability of thrombosis within the VAD by integrating the main factors associated with thrombosis (shear stress, residence time, and coagulation factors) compared to previous studies. There are some limitations in this study. First, this study is solved in a steady method, ignoring the dynamic flow characteristics. This is feasible for continuous flow pumps, and the numerical results in this study are in better agreement with the experimental results. However, for the new generation of pulsed VADs, such as the Lavare cycle pump of the HVAD and the artificial pulse pump of the HeartMate III [59], the transient effects of the pulses need to be considered. Second, platelet activation, adhesion and detachment on the wall were not considered in this study, and the effect of the coagulation cascade and the effect of the artificial material on platelet activation and thrombus formation are also worth considering, which will be performed in our future work.

## 5. Conclusions

In this study, a more accurate model for thrombus assessment has been constructed by integrating parameters such as shear stress, residence time and coagulant factors, so as to accurately assess the probability of thrombosis in VAD patients. A mathematical model was employed to assess platelet activation and thrombosis within the VAD. By solving the transport equation, the influence of various factors such as shear stress, residence time and coagulation factors on platelet activation were investigated. The diffusion equation was applied to determine the role of activated platelets and substance deposition on thrombus formation. The momentum equation was introduced to describe the obstruction to blood flow when thrombus is formed, and finally a more comprehensive and accurate model for thrombus assessment in patients with VAD was obtained. Numerical simulations of three clinically used VADs (CH-VAD, HVAD and HMII) were performed using this model. It was found that the percentage of activated platelets within the VAD obtained by solving the thrombosis model developed in this study was in high agreement with the experimental data (r² = 0.984), the likelihood of thrombosis in the regions of the simulation showed excellent correlation with the clinical statistics (r² = 0.994), and the regions of high thrombotic risk predicted by the simulation were consistent with the experimental results. Further study revealed that the three clinical VADs (CH-VAD, HVAD and HMII) were prone to thrombus formation in the inner side of the secondary flow passage, the clearance between cone and impeller, and the corner region of the inlet pipe, respectively. The risk of platelet activation and thrombus formation for the three VADs was low to high for CH-VAD, HVAD, and HMII, respectively. Simulation results of thrombotic risk received with this model showed excellent correlation with experimental and clinical data. It is important for determining the degree of platelet activation in VAD and identifying regions prone to thrombus formation, as well as guiding the optimal design of VAD and clinical treatment.

## Figures and Tables

**Figure 1 bioengineering-09-00235-f001:**
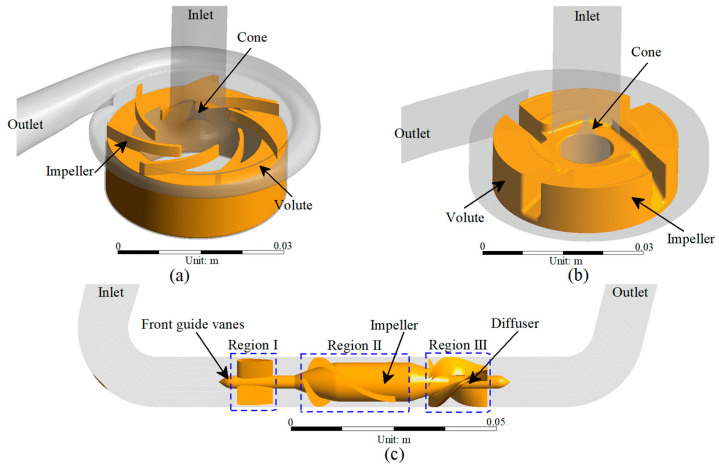
3D model of the studied VADs: (**a**) CH-VAD; (**b**) HVAD; and (**c**) HM II.

**Figure 2 bioengineering-09-00235-f002:**
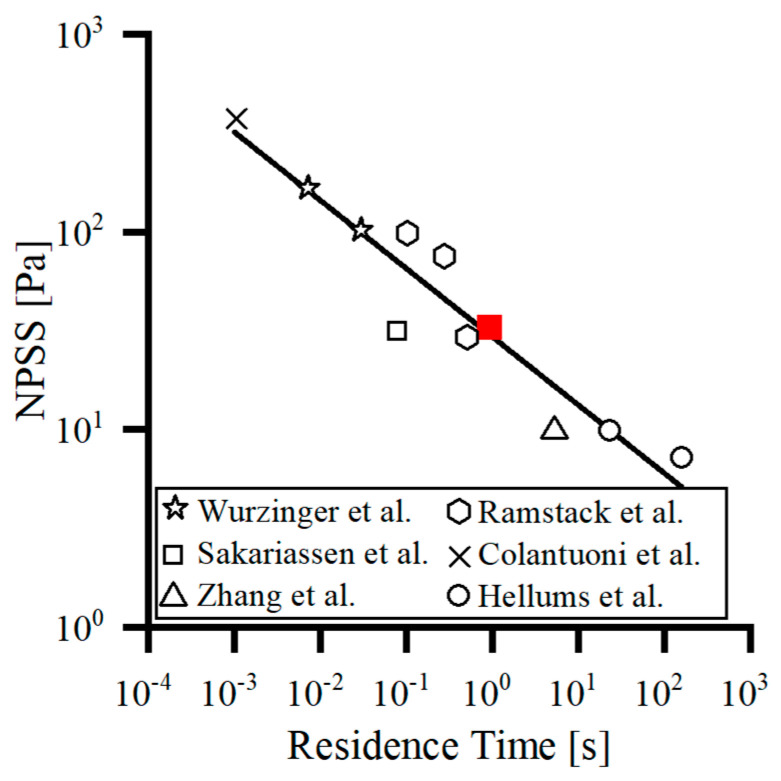
The threshold curve for platelet activation induced by NPSS-RT was fitted by Fraser et al. [42] based on previous studies [43,44,45,46,47].

**Figure 3 bioengineering-09-00235-f003:**
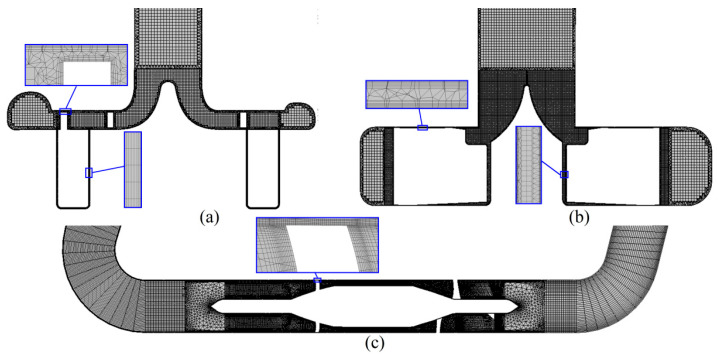
Mesh of the studied VADs: (**a**) CH-VAD; (**b**) HVAD; and (**c**) HM II.

**Figure 4 bioengineering-09-00235-f004:**
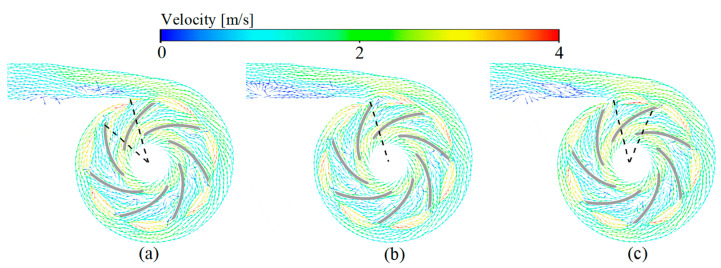
Rotor position independent verification of the CH-VAD: (**a**) Base position −20°; (**b**) Base position; and (**c**) Base position +20°.

**Figure 5 bioengineering-09-00235-f005:**
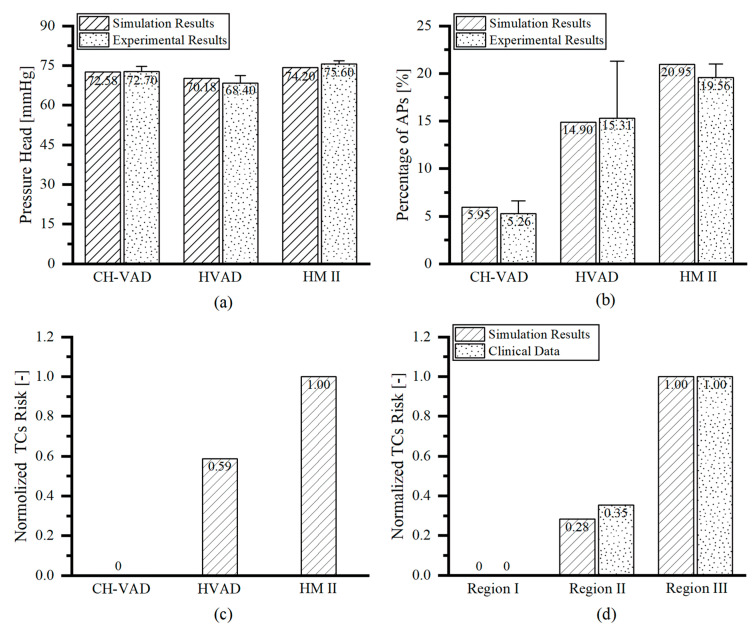
VAD hemocompatibility assessment and comparison for CH-VAD, HVAD and HMII: (**a**) Pressure head (correlation coefficient with experimental results [26] r^2^ = 0.999); (**b**) Platelet activation ratio (correlation coefficient with experimental results [26] r^2^ = 0.984); (**c**) Thrombotic risk; and (**d**) Assessment of thrombotic risk in different regions of HMII (correlation coefficient with clinical data [53] r^2^ = 0.994).

**Figure 6 bioengineering-09-00235-f006:**
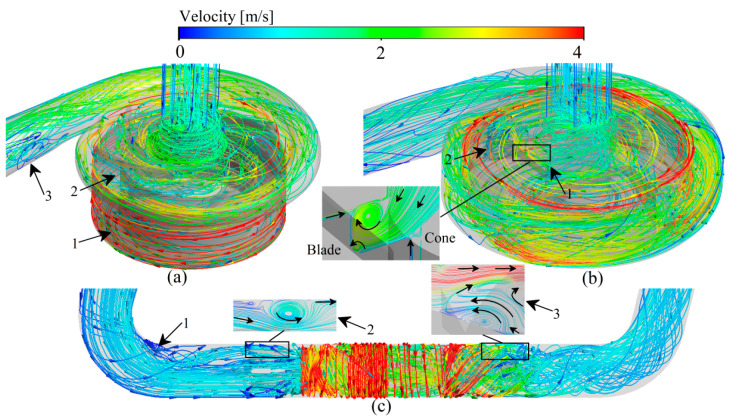
Typical streamlines of the relative velocity predication of three studied VADs: (**a**) CH-VAD; (**b**) HVAD; (**c**) HMII.

**Figure 7 bioengineering-09-00235-f007:**
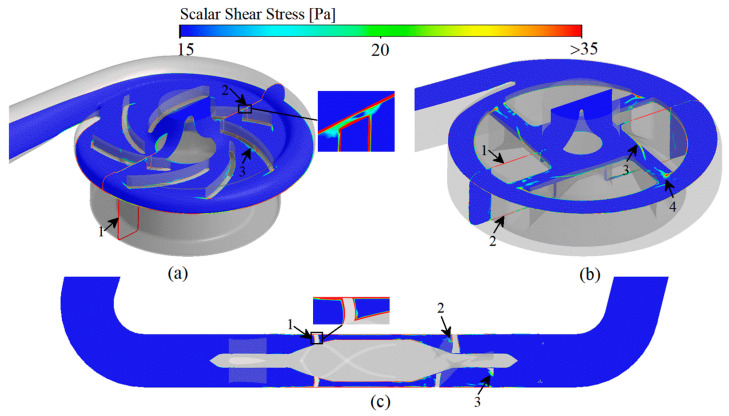
Scalar shear stress prediction of three studied VADs: (**a**) CH-VAD; (b) HVAD; (**c**) HMII.

**Figure 8 bioengineering-09-00235-f008:**
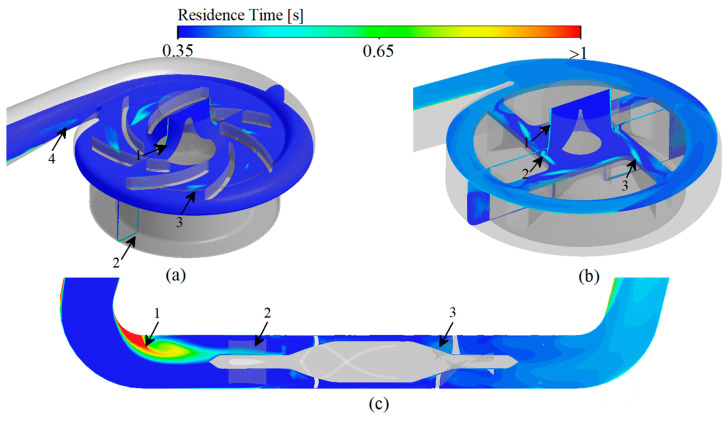
Residence time prediction of three studied VADs: (**a**) CH-VAD; (**b**) HVAD; (**c**) HMII.

**Figure 9 bioengineering-09-00235-f009:**
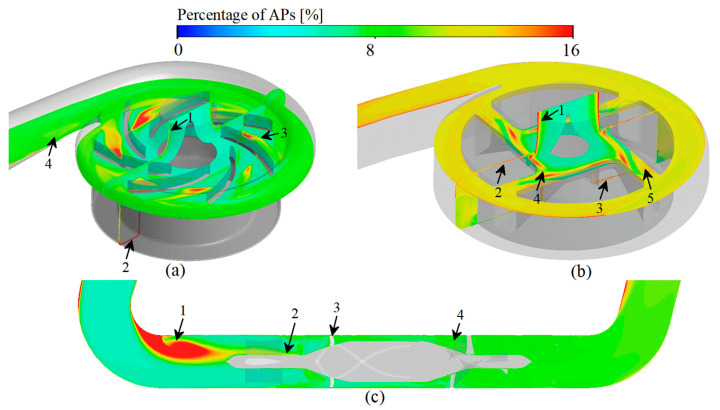
Activated platelets prediction of three studied VADs: (**a**) CH-VAD; (**b**) HVAD; (**c**) HMII.

**Figure 10 bioengineering-09-00235-f010:**
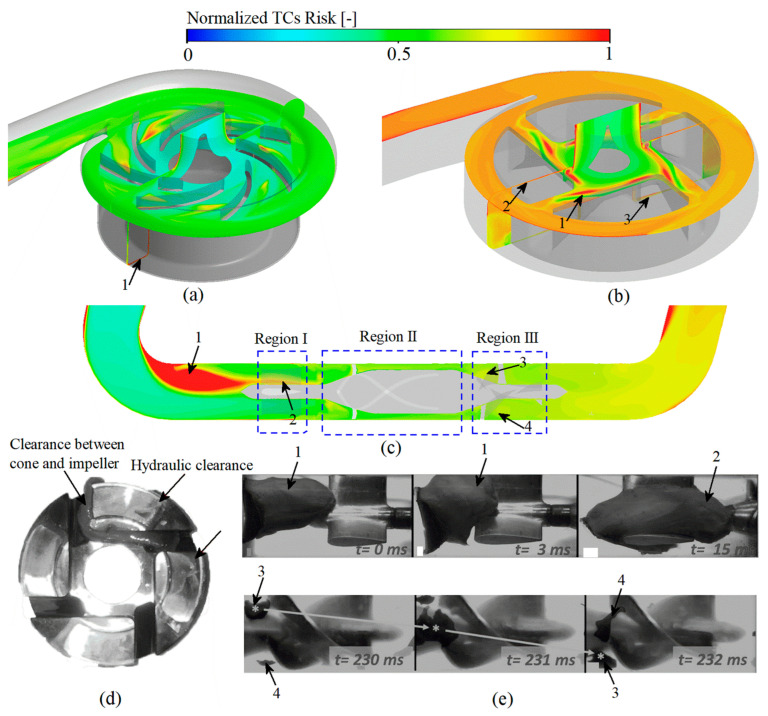
High thrombosis risk regions predicted of the studied VADs: (**a**) Simulation prediction of CH-VAD; (**b**) Simulation prediction of HVAD; (**c**) Simulation prediction of HM II; (**d**) Experimental prediction of HVAD, this picture is reproduced from the study performed by Schalit et al. [4]); (**e**) Experimental prediction of HM II [55].

**Table 1 bioengineering-09-00235-t001:** The thresholds of the reaction variable.

Variable	RT_0_ (t_0_)	NPSS_0_ [27]	γ_0_ [27]	CF_0_ [27]	TC_0_ [27]
Threshold	1 s	35 Pa	100/s	10 nmol	200 nmol

**Table 2 bioengineering-09-00235-t002:** The boundary conditions of the reaction variable.

Variable	RT	APs [15]	RPs [15]	CFs [38]	TCs
Inlet Boundary	0 s	25 × 10^12^/m^3^	475 × 10^12^/m^3^	0 mol	0 mol
Outlet Boundary	Flux 0	Flux 0	Flux 0	Flux 0	Flux 0
Wall Boundary	Flux 0	Flux 0	Flux 0	Flux 1	Flux 0

## Data Availability

The study does not report any data.

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
