# Peer review of "A New Mathematical Numerical Model to Evaluate the Risk of Thrombosis in Three Clinical Ventricular Assist Devices"

_bioengineering, 2022, doi:10.3390/bioengineering9060235_

Round 1
Reviewer 1 Report
In this work, three-dimensional flow of blood is modeled in three different VADs using CFD and models for thrombus assessment have been implemented.
- Dimensions of the geometries shown in the article should be provided. Alternatively, CAD geometries can be uploaded as a supplement material.
- CFD model is described after the discussion of RT, NPSS and other models. In fact, all the models presented in Sections 2.2 and 2.3 require velocity which would be obtained from CFD by solving mass and momentum conservation equations. Therefore, section 2.4 should be described. Further, it should be clearly pointed out how different models integrate.
- What are the boundary conditions for RT, RP, AP and TC?
- Equation 2 presents NPSS. How is it different from SSS? The formula for shear rate tensor given just after Eq. 2 is not correct. Probably, it is a typo. Are the simulations performed at steady state or are they transient?
- This work takes the models for RT, RP, AP and TC from literature. The novelty of the work is not exactly clear! Does it lie in employing these models for the analysis of the three VADs?
- The flow description considered in this work is Eulerian. However, the use of term ‘Eulerian’ gave a first impression that the work may involve multiphase Eulerian-Eulerian approach to model the blood which is not true. Therefore, I suggest avoid using the word Eulerian.
Author Response
Responses to the Comments of reviewer1
We would like to take this opportunity to thank the reviewers for their thoughtful and constructive reviews of our manuscript and their appreciation of our work. The comments and suggestions of the reviewers have helped us improve the manuscript. In the revised manuscript, the revised texts were highlighted in red to address the comments of Reviewer1. The below are our point-to-point responses to the comments and suggestions by the reviewer 1. The original review comments are in italic format and the corresponding response follows each comment.

Reviewer 2 Report
Manuscript ID: bioengineering-1708635
Type of manuscript: Article
Title: A New Mathematical Model to Evaluate the Risk of Thrombosis in Three
Clinical Ventricular Assist Devices
- Introduction:
The aim of this article is to propose a numerical model to predict the risk of formation of a thrombus for patient supported with a ventricular assist devices (VAD). The authors modeled resting platelets (RPs), activated platelets (APs) and coagulation factors (CFs) as dilution chemicals passively transported with the blood stream. For this purpose, the authors introduced equations to model these factors, in a standard way, using the CEL language macro of the ANSYS CFX software. The authors compared the results obtained and the risks of thrombus formation for 3 types of VAD. This subject is a crucial issue to improve the knowledge of the thrombus risks of a patient with VAD and a better follow-up of these patients. However, this article raises some comments.
- General Comments:
First of all, I think the following title would be more appropriate : A New Mathematical Numerical (or Finite Element) Model to Evaluate the Risk of Thrombosis in Three Clinical Ventricular Assist Devices
The authors cited several papers providing different values for abnormal shear stress and residence time. Many papers provided low and high thresholds for shear stresses. In this paper a threshold of 35 MPa is given in Table 1. This choice must be justified as well as the absence of a low threshold for shear stress.
In the article, the authors often describe the risk zones as those where the shear stress is high. This needs to be more detailed and needs to be compared to the threshold selected. (a zone where the shear stress is high but significantly lower than the threshold, does not mean it is a risk zone).
It should be noted that the solving of the problem (using Ansys CFX) is relatively conventional and does not involve a very innovative approach. The use of CEL macros is currently quite common for the solving of multi physic problems using Ansys CFX software.
Section : 2.1. Studied VADs and geometries
Some dimensions are needed in the sketches in figure 1.
Section : 2.4. CFD Methods
Authors must specify the version of Ansys used.
One point needs to be clarified. Thus, the authors write: … During the simulation, all these three pump speeds were chosen according to their H-Q curve (3000rpm for CHVAD, 2800rpm for HVAD and 9200rpm for HMII, respectively)…
(H-Q curve is not defined)
However, in the following of the article and among others, in the results sections, no time notion is mentioned. It seems then that the calculations are made for a stationary configuration chosen without justification (position of the impeller). The authors must clarify this important point.
Figure 4 (like the following ones) represents the velocity field for an arbitrary position of the impellers. Could the authors confirm?
A precise justification of this point is required.
The boundary conditions need some details. Only an average inlet and an outlet flow are imposed? Is the inlet velocity profile correct?
Could the authors give an approximate time of solving on a specified type of computer?
Section : 2.5. Mesh Details and Sensitivity Analysis
The average mesh size must be specified.
It seems that there is no fluid/structure interaction. This assumption must be justified.
Section : 3.1. Hemodynamic results and simulation results validation
For a clearer understanding, the values of the experimental results should be shown in the text or in figure 3.
Could the authors explain how normalized thrombosis risk is calculated and how the clinical values are assessed?
Section : 3.2. Velocity fields
Could the authors explain if the turbulence in area 3 (Figure 4-a) is physical and not related to the vicinity of the outlet where the boundary conditions are applied ?
Section : 3.2. Scalar shear stress and Residence time
First of all, there are 2 sections 3.2!!!
it would be more accurate to speak of shear stress above a threshold rather than high shear stress.
Figure 5 could only represent the interesting zones, i.e. those where the shear stress is higher than the threshold retained in this study.
Section : 3.3. Activated platelets
First of all, this section should be 3.4… and so on (for the following sections).
Figure 5 could only represent the interesting zones, i.e. those where the shear stress is higher than the threshold retained in this study.
Section : 3.4. Thrombosis risk assessment
First of all, this section should be 3.5… and so on (for the following sections).
This section needs to be improved as it is rather confusing and difficult to understand. For example, there is confusion when the authors refer to Figure 8.
For figure 8, is figure 8.d interesting? There is some confusion in the legend...
Section : 4. Discussion
In this chapter, a more substantial justification of the results and a better explanation of the numerical results would be interesting. These explanations could be an interesting alternative to some of the (too numerous) critical comments on the existing models.
- Conclusion
Due to the different comments, this interesting work must be completed and rewrite. In the present form, this article cannot be published in Bioengineering. This paper required a major revision before to reconsidered.
I would like to thank the authors for writing this article, and the editor for inviting me to review the article. It was a very interesting task.

Author Response
Responses to the Comments of reviewer2
We would like to take this opportunity to thank the reviewers for their thoughtful and constructive reviews of our manuscript and their appreciation of our work. The comments and suggestions of the reviewers have helped us improve the manuscript. In the revised manuscript, the revised texts were highlighted in blue to address the comments of Reviewer 2. The below are our point-to-point responses to the comments and suggestions by the reviewer 2. The original review comments are in italic format and the corresponding response follows each comment.

Reviewer 3 Report
Please see attached pdf file.

Author Response
Responses to the Comments of reviewer3
We would like to take this opportunity to thank the reviewers for their thoughtful and constructive reviews of our manuscript and their appreciation of our work. The comments and suggestions of the reviewers have helped us improve the manuscript. In the revised manuscript, the revised texts were highlighted in purple to address the comments of Reviewer 3. The below are our point-to-point responses to the comments and suggestions by the reviewer 3. The original review comments are in italic format and the corresponding response follows each comment.

Round 2
Reviewer 2 Report
This new version has been significantly improved and can be published in the journal Bioengineering.
Change the font for figure 8
Reviewer 3 Report
The authors exhaustively addressed reviewer's issues, essentially on the basis of documented literature and common practice. I think that the proposed study deserves scientific consideration. Therefore, the recommendation is: accept in present form.